# Long-Term Corrosion Behavior of Strong and Ductile High Mn-Low Cr Steel in Acidic Aqueous Environments

**DOI:** 10.3390/ma15051746

**Published:** 2022-02-25

**Authors:** Jin Sung Park, Si On Kim, Young Jae Jeong, Soon Gi Lee, Jong Kyo Choi, Sung Jin Kim

**Affiliations:** 1Department of Advanced Materials and Metallurgical Engineering, Sunchon National University, Jungang-ro, Suncheon 540-742, Korea; pjs1352@naver.com (J.S.P.); kzo1102@naver.com (S.O.K.); zzz_8425@naver.com (Y.J.J.); 2POSCO Technical Research Laboratories, Gyungbuk, Pohang 790-704, Korea; sjhrte1156@nate.com (S.G.L.); pogksj@naver.com (J.K.C.)

**Keywords:** high manganese steel, corrosion, polarization, impedance, scale, acidic environment

## Abstract

To expand the industrial applicability of strong and ductile high Mn-Low Cr steel, a deeper understanding and mechanistic interpretation of long-term corrosion behavior under harsher environmental conditions are needed. From this perspective, the long-term corrosion behaviors of 24Mn3Cr steel under acidic aqueous conditions were examined through a comparison with conventional ferritic steels using the electrochemical measurements (linear polarization resistance and impedance), and immersion test followed by the metallographic observation of corrosion morphologies. In contrast to conventional ferritic steels, 24Mn3Cr steel, which had the lowest corrosion resistance at the early immersion stages (i.e., the highest corrosion current density (*i_corr_*) and lowest polarization resistance (*R_p_*)), showed a gradual increase in corrosion resistance with prolonged immersion. Owing to the slow formation kinetics of (Fe,Cr)-enriched oxide scale, a longer incubation time for ensuring a comparatively higher corrosion resistance is required. On the other hand, conventional ferritic steels had an oxide scale with less densification and a lower elemental enrichment level that did not provide an effective anti-corrosion function. From the results, this study can provide significant insight into the industrial applicability of the high Mn-low Cr steel by providing the mechanistic interpretation of corrosion behaviors in acidic aqueous environments.

## 1. Introduction

Recently, the development of ternary-based austenitic steel alloys (Fe_1−x−y_Mn_x_Cr_y_ with y << x << 1 and lower C concentrations) with attractive mechanical properties, an inhibiting nature in relation to hydrogen penetration, and erosion–corrosion resistance, were reported [1,2,3]. Despite the intrinsic weakness arising from their higher Mn concentrations (i.e., the standard reduction potential of Mn (−1.18 V) is much lower than that of Fe (−0.447 V) [4,5]), their long-term corrosion resistance is much higher than conventional API (American petroleum institute)-grade low-C steel and is even comparable to 9% Ni steel in a neutral aqueous environment. A previous study [1] revealed that their superior resistance to the dynamic flowing erosion–corrosion as well as static corrosion was attributed primarily to the formation of an (Fe,Cr)-enriched oxide scale (i.e., α-Fe_2_-xCr_x_O_3_) with an inhibiting nature on the surface. In 9% Ni steel, the higher corrosion resistance is caused mainly by the formation of an oxide scale composed of Ni_x_Fe_3_-xO_4_ and NiO/Ni_2_O_3_ on the surface [1,6,7]. However, these surface characteristics can only be manifested under a neutral aqueous environment. The corrosion study with high Mn-bearing steel (i.e., 18Mn3Cr steel) conducted by Park et al. [8] was based on a sweet environment. They reported that the addition of Cr to high Mn steel improved the resistance to CO_2_ corrosion owing to the increase in the stability of the corrosion products formed on the steel surface. In addition, Kim et al. [9] reported the beneficial effects of the addition of Cr and Mn to high Mn steel (i.e., 18Mn and 18Mn5Cr steel) on the corrosion resistance in a 3.5% NaCl solution at 60 °C. They showed that the alloying elements of Mn and Cr in high Mn steel contributed to the increase in goethite (α-FeOOH) in non-adherent rust (NARs) and to the refinement of the adherent rust (AR) grain. However, they focused only on the corrosion resistance of high Mn steel in a neutral environment. J. Bosch et al. [10] reported the corrosion behavior of high Mn-bearing steel (i.e., 22.6–30 Mn steel) under a variety of aqueous conditions (i.e., neutral, and alkaline environments), but the mechanistic interpretation of the corrosion and scale forming processes was limited. Hence, to expand the applicability of 24% Mn-3% Cr bearing steel to various industrial sectors, it is necessary to advance the knowledge of the long-term corrosion behavior under harsher environmental conditions. In particular, the corrosion behaviors and scale forming processes of 24Mn-3Cr steel in the acidic environments, where the anodic dissolution rate could be much higher than in the neutral conditions, needs deeper investigation.

This work extends a previous study to examine the long-term corrosion behavior of 24% Mn-3% Cr bearing steel in comparison with two conventional ferritic steel samples (9Ni and API-grade low-C steels) under exposure to acidic environments. For this, the change in the electrochemical polarization and impedance behaviors of the steels with an increasing immersion in an acidic solution was interpreted based on the characteristics of the corrosion scale formed on their surfaces.

## 2. Materials and Methods

### 2.1. Materials Information

The tested materials under current investigation were a high Mn (24%)-bearing steel with a face-centered cubic (FCC) structure and two commercial structural steel samples: 9% Ni steel and API-grade X70 steel, with a body-centered cubic (BCC) structure. Their chemical compositions are listed in Table 1, and they are called ‘24Mn3Cr’, ‘9Ni’, and ‘APIX70’, respectively, hereinafter. The mechanical properties and detailed manufacturing processes for the tested samples have been reported elsewhere [1]. Before testing, all samples (2 × 2 × 0.5 cm^3^) were mechanically polished to 2000 grit sandpaper and cleaned ultrasonically in ethanol.

### 2.2. Open Circuit Potential and Linear Polarization Resistance Measurements

The electrochemical experiments were carried out with a conventional three-electrodes cell system consisting of a sample, a Pt grid, and a saturated calomel electrode (SCE), which served as the working, counter, and reference electrode, respectively. The sample fastened to the corrosion cell was then immersed in an acidic 3.5% NaCl solution adjusted to pH 3 using CH_3_COOH, with an exposure area of 1 cm^2^, for 42 days. The tested samples were considered as materials for a slurry pipeline that should meet the NACE standard (i.e., NACE TM0284 [11]). Considering that the acetic acid was used for the tested solution in the NACE standard, the pH was adjusted using acetic acid. The change in the open circuit potential (OCP) with an increasing immersion time was plotted continuously. For the linear polarization resistance (LPR) measurements, the potential from −20 mV to 20 mV vs. the OCP of the sample was applied using a potentiostat (Gamry Reference 600, Pennsylvania, U.S.). The corrosion current density (*i_corr_*) was quantified by fitting the experimental LPR data to the theoretical model based on the Wagner–Traud equation [12] described below.
(1)$$i=i_{corr}\left[\exp\left(\frac{2.303\left(E-E_{corr}\right)}{\beta_{a}}\right)-\exp\left(\frac{-2.303\left(E-E_{corr}\right)}{\beta_{c}}\right)\right]$$
where *i*, *i_corr_*, *β_a_*, *β_c_*, *E*, and *E_corr_* are the measured total current density, corrosion current density, anodic Tafel slope, cathodic Tafel slope, measured potential, and corrosion potential, respectively.

The changes in *i_corr_* and *mpy* (i.e., mils per year, which was derived from *i_corr_* by the unit conversion [13]) with an increasing immersion time were obtained on the same sample. The experiment was performed three times to ensure reproducibility, and the mean values and their standard deviations are listed in Table 2.

### 2.3. Electrochemical Impedance Measurements

Electrochemical impedance spectroscopy (EIS) was performed over the frequency range, from 100 kHz to 10 mHz, with a sinusoidal AC potential of ±10 mV at the OCP. The three-electrodes cell system and sample preparation processes were the same as those for the LPR experiment, as described above. The Nyquist plots obtained were fitted with an equivalent circuit model using the Gamry Echem Analyst software (ver. 7.07), and the fitted curves and parameters were plotted. The experiment was performed three times to ensure reproducibility, and the mean values and their standard deviations were listed in Table 3.

### 2.4. Analysis of the Corrosion Scale

After immersion in an acidic solution for 7 and 28 days, the surface and cross-section of the samples were observed using field-emission scanning electron microscopy (FE-SEM) (Hitachi, Tokyo, Japan). Prior to the cross-sectional observation, the samples, which had been immersed in the solution, were mounted, grounded mechanically with sandpaper, and micro-polished with a 1 μm diamond suspension. The elemental compositions in the corrosion scale formed on the sample surface were examined by energy dispersive spectroscopy (EDS) (Bruker, Berlin, Germany).

## 3. Results and Discussion

### 3.1. Electrochemical Corrosion Behaviors

Similar to the case of neutral aqueous conditions reported previously [1], the OCPs of the 24Mn3Cr sample were comparatively lower than the other two samples over the entire immersion period, as shown in Figure 1. Considering that the standard reduction potential, *E*_0_, of Mn (i.e., Mn = Mn^2+^ + 2e^−^), is approximately −1.18 V, which is much lower than that of Fe (−0.447 V) and Ni (−0.257 V) [4,14], the addition of 24 wt. % Mn to steel decreased the OCP of steel at the early stages of immersion in aqueous conditions. The OCPs of 24Mn3Cr were ennobled slightly with an increasing immersion time. On the other hand, the much higher OCPs of 9Ni and APIX70 at the early stages decreased gradually with an increasing immersion time. In particular, marked decreases in the OCP of 9Ni were notable. Although OCP evolution may be associated with the characteristics of the corrosion scale formed on the steel samples, it cannot be a direct indication of the corrosion kinetics, and the polarization behaviors need to be examined further.

Figure 2a–c present the curve fitting to LPR data in a semi-log format. In addition, Table 2 lists the fitted parameters of the LPR data evaluated in an acidic solution. The *E_corr_* values of APIX70 and 9Ni samples decreased in the active direction with prolonged immersion, whereas the *E_corr_* of the 24Mn3Cr sample showed a tendency to continuously increase in the noble direction. Compared to the case of a neutral environment [1], the *i_corr_* of the samples measured in an acidic solution, which are listed in Table 2, were considerably higher, presumably due to the higher cathodic reduction rate by H^+^ (4H^+^ + O_2_ + 4e^−^ → 2H_2_O) in the aerated solution and the thermodynamic instability of the corrosion scale, composed mostly of metal oxides/hydroxides, formed on the steel samples. In particular, the *i_corr_* of 24Mn3Cr measured at one-day immersion was significantly higher (i.e., more than five times compared to that of 9Ni). In addition, it was notable that the differences in *i_corr_* among the samples within one day of immersion under acidic conditions (more than 100 μA/cm^2^) were much greater than those under neutral conditions (approximately 10 μA/cm^2^ [1]). These suggest that the surface of the 24Mn3Cr sample showed more rapid dissolution kinetics of metallic cations (Mn^2+^/Fe^2+^/Fe^3+^) at the initial stages under acidic conditions. Interestingly, from Figure 2d (i.e., *i_corr_*–immersion time chart), considerable differences between the 24Mn3Cr and 9Ni samples were observed with an increasing immersion time. In contrast to the 9Ni sample, the corrosion resistance of the 24Mn3Cr sample was improved significantly after a prolonged immersion, as evidenced by the markedly large decline in *i_corr_*. The order of *i_corr_* between the 9Ni and 24Mn3Cr samples measured initially was reversed after 14–21 days of immersion. 

This trend was supported further by the change in charge transfer (*R_ct_*) and scale resistances (*R_scale_*) obtained by fitting to the Nyquist plots of EIS with immersion time (Figure 3). Except for the very early immersion stages, the oxide layers were formed on the steel surface with a prolonged immersion and the two time constants were contained in an equivalent circuit to fit the Nyquist plots. In contrast to the bare steel, the corrosion scale formed on the steel surface could be measured as another resistance due mainly to its semi-conducting properties [15]. In this regard, the Nyquist plots with prolonged immersion except for one day of immersion were fitted with an equivalent circuit including two time constants. The two types of equivalent circuit models used in this work are shown in Figure 3. 

The fitted parameters from the curve fitting to Nyquist plots with the equivalent circuit models are listed in Table 3. As shown in the *R_p_* (*R_ct_* + *R_scale_*)–immersion time charts, the lowest *R_p_* of 24Mn3Cr sample at the initial stages increased considerably with an increasing immersion time, showing the highest value, but it was vice versa in the case of the 9Ni sample. The highest *Q_dl_* of the 24Mn3Cr sample was within 1–24 h of the immersion, which suggests that the formation of a scale by the adsorption of metallic ions [16,17,18] at the sample surface was insignificant. As the immersion time was increased, however, the formation of a scale with an inhibiting nature (i.e., higher *R_p_*, which can be found in Table 3) on the surface of the 24Mn3Cr sample was expected. In particular, as shown in Table 3, the *R_scale_* of 24Mn3Cr was the highest compared to the other samples, and it showed a tendency to increase significantly with a prolonged immersion. The impedance can be briefly expressed in the following equation [19,20,21,22]:(2)$$Z=\frac{1}{Y_{0}}{\left[{\left(j\omega \right)}^{n}\right]}^{-1}$$
where *Y*_0_ is the magnitude of the CPE, *j* is the imaginary number, *w* is the angular frequency, and *n* is the depression parameter (0 < *n* <1).

Because the derived values of *n* were less than 1 (see Table 3), this system at the interface deviated from an ideal capacitance, and the depression angle (α) can be quantified as follows [23,24].
(3)$$\alpha =\frac{2\left(1-n\right)}{\pi }$$

Considering that the increase in *α* (derived from *n_scale_*) could be ascribed to the heterogeneity of the surface by increasing the surface roughness and forming the porous layers [24,25], a much smaller α (APIX70: 0.116, 9Ni: 0.088, and 24Mn3Cr: 0.076) for the 24Mn3Cr sample with a prolonged immersion time could also suggest that the sample may have had surface inhibiting properties with a comparatively higher level of homogeneity and a smaller level of porosity. An interesting point which is different from the measurements in a neutral condition reported previously [1] is that a longer incubation time was required to ensure the comparatively higher corrosion resistance of the 24Mn3Cr sample in an acidic environment. On the other hand, the 9Ni sample, which showed superior corrosion resistance in a neutral solution, exhibited a significant deterioration of the corrosion resistance as the immersion time was increased in an acidic environment. The 9Ni sample showed the lowest corrosion resistance among the three samples tested after approximately 14–21 days of immersion.

### 3.2. Characteristics of the Corrosion Scales

The morphological and compositional differences of the corrosion scale formed on the 24Mn3Cr, 9Ni, and APIX70 samples were compared, and they are shown in Figure 4, Figure 5, Figure 6 and Figure 7. As shown in Figure 4, there was no pre-existing corrosion damage or corrosion scale on the steel surface before immersion. At an early immersion stage (seven days), the sample surface was not covered uniformly with the (Fe,Cr)-enriched oxide (Table 4), which was characterized previously as α-Fe_2−x_Cr_x_O_3_, [1]. This is clearly different from the results obtained under a neutral environment, which showed rapid formation kinetics of the corrosion scale [1]. At the early immersion stages, the much higher *i_corr_*, and lower *R_ct_* and *R_scale_* of the 24Mn3Cr sample, presented in the previous section, can be understood by the slower formation kinetics of α-Fe_2−x_Cr_x_O_3_, known as an effective barrier against further corrosion [1,26]. 

With a prolonged immersion (28 days, Figure 6 and Table 5), however, the sample surface was fully covered with the inner layer of the (Fe,Cr)-enriched scale. The outer layer of the Fe(Mn)-based oxide scale was not stably formed even at this stage. These results suggest that a longer incubation time is required for the stable formation of a corrosion scale that can act as a barrier against further corrosion in acidic environments. The amounts of C and Cl, detected by EDS analysis, were very insignificant in this study, which may be ascribed to the limited resolution of EDS for light elements, such as C and Cl. In this case, the use of analysis techniques with a higher resolution for light elements, such as electron probe microanalysis (EPMA), would be needed.

On the other hand, the surface of the 9Ni sample was mostly covered with an Fe(Ni)-based oxide scale at the early stages of immersion (within seven days). Although there was an even interface between the scale and the underlying matrix, the covering ratio by the scale was comparatively higher than the case of the 24Mn3Cr sample. With a prolonged immersion to 28 days, the scale became much thicker, which was more than eight times the thickness measured in a neutral environment [1]. On the other hand, the densification and Ni-enriched level in the scale were somewhat lowered, considering the elemental distribution map and point analysis of Fe/Ni/O. In particular, the outermost layer in the scale was not covered with a Ni-enriched oxide (i.e., NiO) formed under neutral conditions [1]. This can be understood by considering the significant decrease in the thermodynamic stability of Ni-based oxides and the increase in Ni^2+^ activity with the increasing acidity (refer to Pourbaix diagram of Ni [27,28]). The decrease in the corrosion resistance of 9Ni with an increasing immersion time in this acidic solution, based on the gradual increase in *i_corr_*, and the decrease in *R_p_* can be understood in this way. In the case of the APIX70 sample, there was no significant difference in the elemental composition in the scale except that the scale became much thicker than that under the neutral conditions [1].

### 3.3. Corrosion Mechanism of High Mn-Low Cr Steel

Based on the electrochemical analyses and phenomenological observations, the schematic diagram showing the long-term corrosion mechanism of 24Mn3Cr steel under acidic environments is provided in Figure 8. At the initial corrosion stage, the rapid dissolution of Mn on the surface occurred due to the lower standard reduction potential of Mn (−1.18 V) than that of Fe (−0.447 V). Although the steel surface was not covered uniformly with the (Fe,Cr)-enriched oxide at the early corrosion stage, a dense and stable (Fe,Cr)-enriched oxide layer was gradually formed on the steel surface with prolonged immersion, providing anti-corrosion properties.

In short, the corrosion inhibiting ability of the ternary (high Mn-low Cr-balanced Fe)-based austenitic steel under long-term exposure to acidic environments is far superior to that of 9Ni, which is considered a more expensive steel alloy. This suggests that the applicability of high Mn-low Cr-based austenitic steel as a structural material can be further expanded to various industrial fields requiring contact with acidic aqueous environments.

## 4. Conclusions

This study examined the long-term corrosion behavior of three tested steels in an acidic environment. The change in the electrochemical corrosion behaviors with immersion time was interpreted from the characteristics of the corrosion scale formed on the steel surface. The major findings are as follows. 

At the early stages of immersion in an acidic solution, the 24Mn3Cr steel sample had a much higher corrosion current density (*i_corr_*) and a lower polarization resistance (*R_p_*) compared to the two ferritic steel samples. In particular, the 9Ni steel sample showed the highest corrosion resistance with the lowest corrosion current and highest polarization resistance. With a prolonged immersion, however, these trends were gradually reversed, and the order of corrosion resistance measured at the later stages of immersion was 9Ni < APIX70 < 24Mn3Cr steel samples. In addition, 24Mn3Cr had the highest scale resistance (*R_scale_*) and the lowest depression angle (α). These results are in contrast to those obtained in a neutral environment reported previously.The formation kinetics of the (Fe,Cr)-enriched oxide scale on the 24Mn3Cr steel sample were much slower under acidic conditions. Considering that the oxide scale can serve as an effective barrier against further corrosion, a longer incubation time is required for the stable formation of the oxide scale and to ensure a comparatively higher corrosion resistance in an acidic environment. On the other hand, the 9Ni steel sample was covered with a thicker (Fe,Ni)-based scale, but the densification and Ni-enriched level in the scale was lowered at later corrosion stages, due presumably to the thermodynamic instability of Ni-based oxides in acidic environments. Furthermore, the APIX70 steel sample was covered only by an Fe-based scale which hardly contributed to the improvement of the corrosion resistance.

## Figures and Tables

**Figure 1 materials-15-01746-f001:**
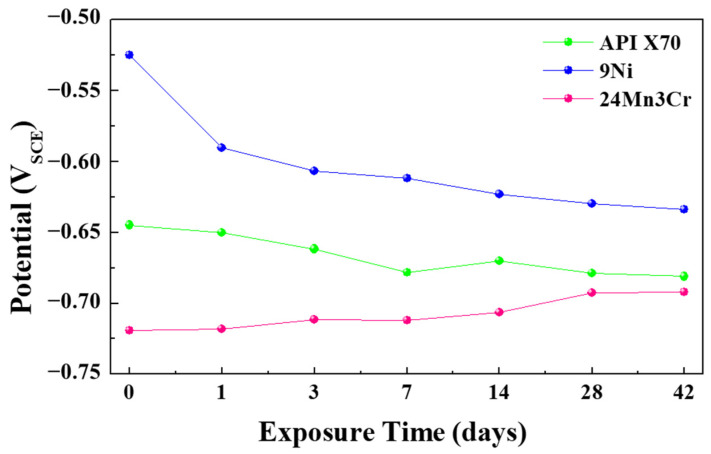
OCP evolution of the three tested samples with an increasing immersion time.

**Figure 2 materials-15-01746-f002:**
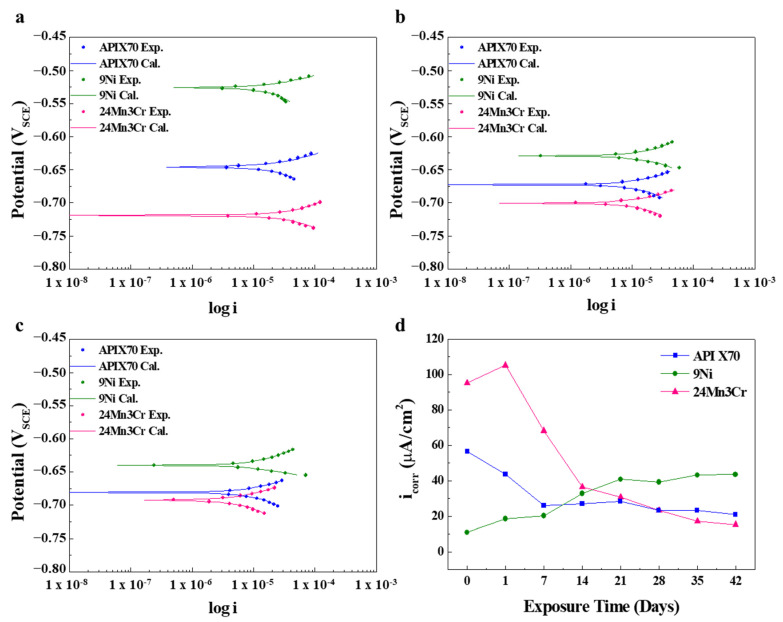
(**a**–**c**) Curve fitting to the LPR curves of the tested samples with different immersion times: (**a**) 1 h, (**b**) 21 days, and (**c**) 42 days; (**d**) Change in *i_corr_* with an increasing immersion time. (Exp. and Cal. are the experimental and fitted data, respectively).

**Figure 3 materials-15-01746-f003:**
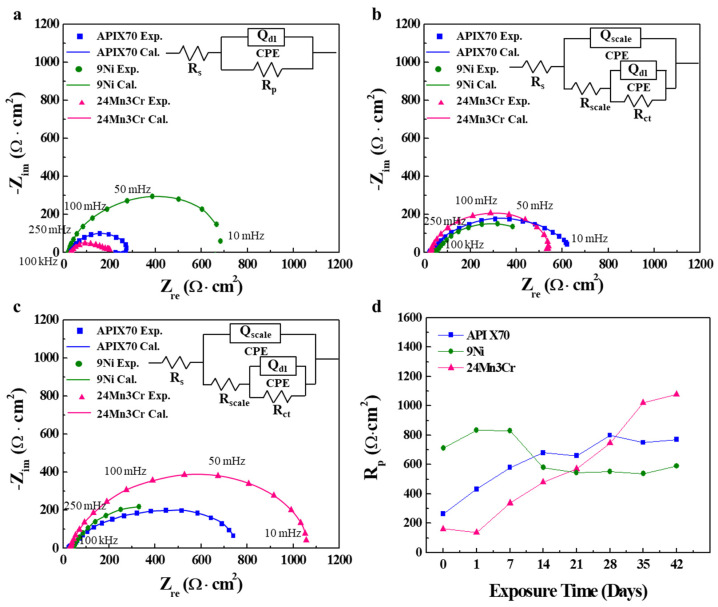
(**a**–**c**) Curve fitting to the Nyquist plots of the tested samples with different immersion times: (**a**) 1 h, (**b**) 21 days, and (**c**) 42 days; (**d**) Change in *R_p_* with an increasing immersion time. (Exp. and Cal. are the experimental and fitted data, respectively).

**Figure 4 materials-15-01746-f004:**
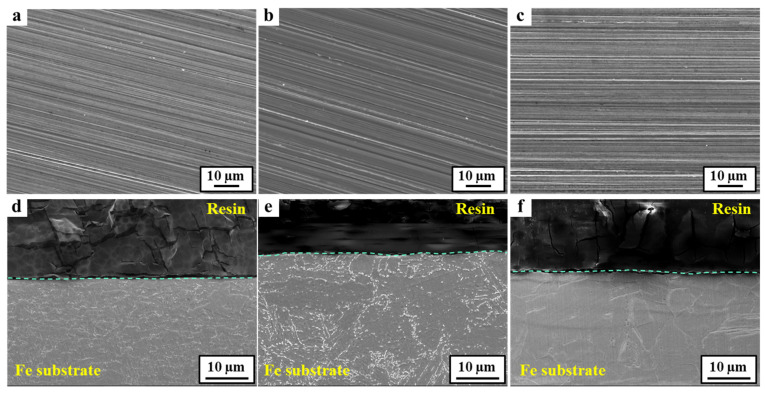
FE-SEM images of the tested samples before immersion. (**a**–**c**) Surface images of APIX70, 9Ni, and 24Mn3Cr, respectively; (**d**–**f**) Cross-sectional images of APIX70, 9Ni, and 24Mn3Cr, respectively.

**Figure 5 materials-15-01746-f005:**
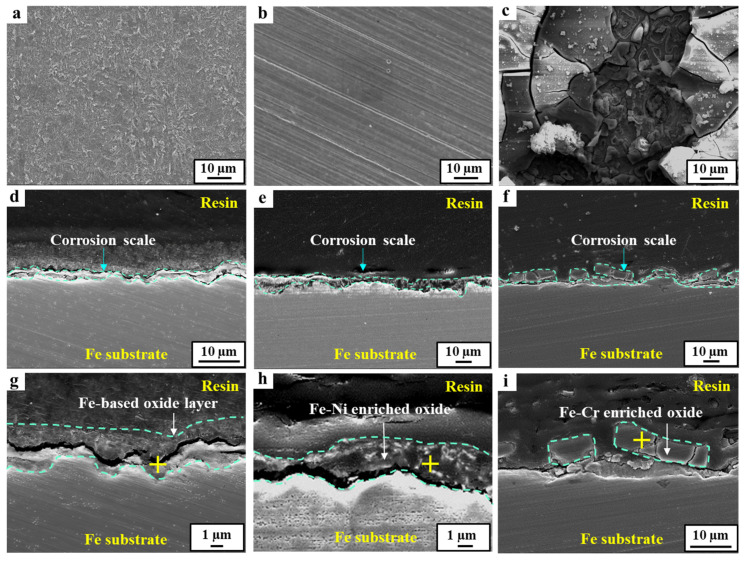
FE-SEM images of the tested samples immersed in an acidic solution for seven days. (**a**–**c**) Surface images of APIX70, 9Ni, and 24Mn3Cr, respectively; (**d**–**f**) Cross-sectional images of APIX70, 9Ni, and 24Mn3Cr, respectively; (**g**–**i**) Magnified view of (**d**–**f**).

**Figure 6 materials-15-01746-f006:**
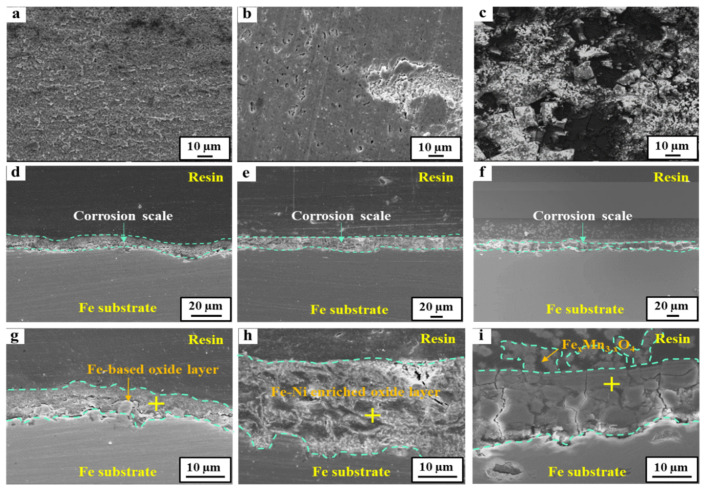
FE-SEM images of the tested samples immersed in an acidic solution for 28 days. (**a**–**c**) Surface images of APIX70, 9Ni, and 24Mn3Cr, respectively; (**d**–**f**) Cross-sectional images of APIX70, 9Ni, and 24Mn3Cr, respectively; (**g**–**i**) Magnified view of (**d**–**f**).

**Figure 7 materials-15-01746-f007:**
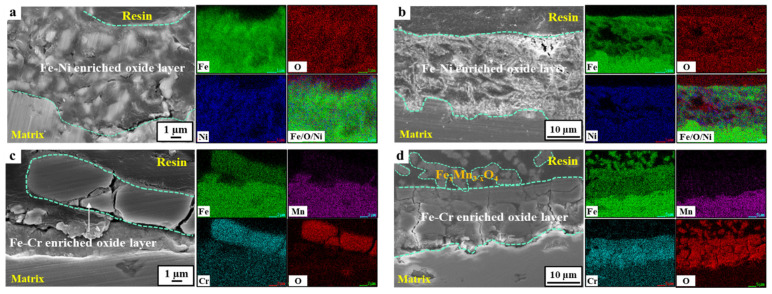
EDS mapping of a cross-section of the tested samples with different immersion times. (**a**,**b**) 9Ni sample immersed for 7 and 28 days, respectively; (**c**,**d**) 24Mn3Cr sample immersed for 7 and 28 days, respectively.

**Figure 8 materials-15-01746-f008:**
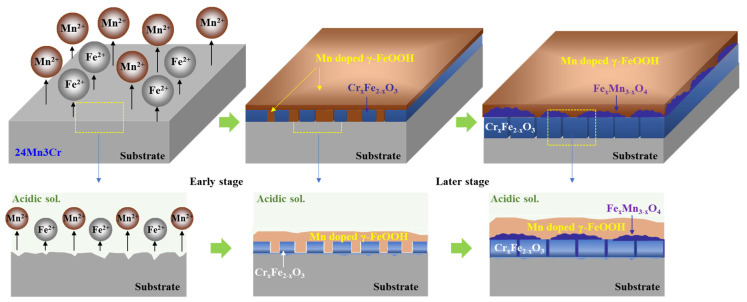
Schematic diagram of the long-term corrosion mechanism of 24Mn3Cr steel under acidic environments.

**Table 1 materials-15-01746-t001:** Chemical composition of the three tested samples.

	C	Mn	Si	Cu	Cr	Ni
24Mn3Cr	0.3–0.4	~24	0.25–0.35	<0.5	3–3.5	0.02–0.03
APIX70	0.01–0.1	1–2	0.2–0.3	<0.05	0.1–0.2	0.01–0.02
9Ni	0.01–0.1	0.6–0.7	0.2–0.3	<0.01	<0.02	~9

**Table 2 materials-15-01746-t002:** Mean values (μ) and their standard deviations (σ) of several fitted parameters obtained by curve fitting to the LPR data of the tested samples.

**APIX70**										
	***i*_corr_ (** **μA·cm^−2^)**		***E_corr_* (** **V)**		***R_p_* (** **Ω·cm^2^)**		***β_a_* (V·Decade^−1^)**		***β_c_* (V·Decade^−1^)**	
**Time**	**μ**	**σ**	**μ**	**σ**	**μ**	**σ**	**μ**	**σ**	**μ**	**σ**
1 h	56.7	2.81	−0.6448	0.008	306.07	9.21	0.03	0.007	0.1	0.001
1 d	43.84	5.11	−0.6501	0.009	392.59	6.55	0.06	0.007	0.12	0.003
7 d	26.03	2.13	−0.6782	0.011	644.81	11.21	0.06	0.006	0.12	0.005
14 d	27.07	1.32	−0.6701	0.016	608.72	8.97	0.06	0.008	0.12	0.001
21 d	28.48	1.85	−0.6725	0.008	580.32	7.25	0.06	0.007	0.12	0.002
28 d	23.31	1.22	−0.6786	0.006	683.28	8.22	0.07	0.006	0.11	0.001
35 d	23.38	3.64	−0.6808	0.005	742.01	15.11	0.07	0.005	0.12	0.002
42 d	21.01	3.45	−0.6762	0.008	782.12	13.58	0.06	0.003	0.12	0.005
**9Ni**										
	***i*_corr_ (** **μA·cm^−2^)**		***E_corr_* (** **V)**		***R_p_* (** **Ω·cm^2^)**		***β_a_* (V·Decade^−1^)**		***β_c_* (V·Decade^−1^)**	
**Time**	**μ**	**σ**	**μ**	**σ**	**μ**	**σ**	**μ**	**σ**	**μ**	**σ**
1 h	11.1	1.01	−0.5249	0.008	356.41	8.12	0.03	0.006	0.12	0.002
1 d	18.64	2.11	−0.5905	0.009	908.38	11.52	0.05	0.008	0.12	0.003
7 d	20.28	1.28	−0.6119	0.011	756.89	16.21	0.11	0.004	0.12	0.001
14 d	32.83	2.32	−0.6228	0.023	461.96	12.08	0.12	0.004	0.12	0.001
21 d	40.87	3.11	−0.6291	0.025	442.86	11.02	0.12	0.008	0.08	0.002
28 d	39.26	6.12	−0.6297	0.018	388.72	10.11	0.12	0.007	0.06	0.006
35 d	43.25	5.44	−0.6294	0.014	343.47	8.89	0.12	0.006	0.05	0.004
42 d	43.57	4.89	−0.6399	0.022	349.22	9.87	0.08	0.005	0.02	0.001
**24Mn3Cr**										
	***i*_corr_ (** **μA·cm^−2^)**		***E_corr_* (** **V)**		***R_p_* (** **Ω·cm^2^)**		***β_a_* (V·Decade^−1^)**		***β_c_* (V·Decade^−1^)**	
**Time**	**μ**	**σ**	**μ**	**σ**	**μ**	**σ**	**μ**	**σ**	**μ**	**σ**
1 h	95.12	6.12	−0.7191	0.012	189.03	5.22	0.09	0.008	0.12	0.002
1 d	105.28	7.55	−0.7179	0.016	158.53	2.11	0.10	0.007	0.12	0.001
7 d	68.24	6.33	−0.7112	0.011	258.72	1.59	0.11	0.009	0.12	0.003
14 d	36.48	1.21	−0.7119	0.009	453.14	7.07	0.06	0.003	0.12	0.005
21 d	30.86	1.23	−0.7004	0.011	541.89	8.55	0.05	0.009	0.11	0.006
28 d	23.3	3.11	−0.6925	0.008	679.79	13.27	0.05	0.009	0.12	0.004
35 d	17.31	2.33	−0.6838	0.013	936.40	20.33	0.05	0.007	0.12	0.004
42 d	15.35	1.28	−0.692	0.009	1158.13	23.12	0.05	0.009	0.11	0.006

**Table 3 materials-15-01746-t003:** Mean values (μ) and their standard deviations (σ) of several fitted parameters obtained by curve fitting to the Nyquist plots of the tested samples.

**APIX70**														
	** *R_s_* ** **(Ω·cm^2^)**		***Q_scale_* (×10^−4 ^** **F·cm^−2^·s^n−1^)**		** *R_scale_* ** **(Ω·cm^2^)**		** *n_scale_* **		***Q_dl_* (×10^−4^** **F·cm^−2^·s^n−1^)**		** *R_ct_* ** **(Ω·cm^2^)**		** *n_dl_* **	
**Time**	**μ**	**σ**	**μ**	**σ**	**μ**	**σ**	**μ**	**σ**	**μ**	**σ**	**μ**	**σ**	**μ**	**σ**
1 h	26.61	1.17	-	-	-	-	-	-	2.45	0.231	263.5	6.51	0.824	0.006
1 d	24.24	1.21	-	-	-	-	-	-	2.82	0.55	433.3	8.21	0.794	0.008
7 d	24.07	1.85	0.88	0.07	32.1	2.87	0.872	0.011	3.54	0.46	557.1	12.17	0.743	0.011
14 d	22.56	2.21	0.44	0.06	34.4	3.54	0.930	0.019	5.51	0.44	642.6	13.22	0.657	0.031
21 d	22.90	2.14	7.01	0.31	159.0	7.22	0.622	0.011	0.87	0.11	474.2	20.19	0.956	0.044
28 d	24.45	4.12	0.27	0.07	19.31	2.87	0.891	0.021	13.01	0.90	892.5	21.36	0.587	0.048
35 d	24.60	3.88	0.78	0.13	19.80	1.23	0.87	0.014	13.26	0.77	800.9	13.18	0.567	0.031
42 d	25.12	2.89	1.79	0.18	18.54	1.36	0.77	0.017	11.13	0.73	821.6	25.13	0.557	0.049
**9Ni**														
	** *R_s_* ** **(Ω·cm^2^)**		***Q_scale_* (×10^−4 ^** **F·cm^−2^·s^n−1^)**		** *R_scale_* ** **(Ω·cm^2^)**		** *n_scale_* **		***Q_dl_* (×10^−4^** **F·cm^−2^·s^n−1^)**		** *R_ct_* ** **(Ω·cm^2^)**		** *n_dl_* **	
**Time**	**μ**	**σ**	**μ**	**σ**	**μ**	**σ**	**μ**	**σ**	**μ**	**σ**	**μ**	**σ**	**μ**	**σ**
1 h	21.05	0.09	-	-	-	-	-	-	1.98	0.31	711.1	11.33	0.854	0.004
1 d	23.41	0.28	-	-	-	-	-	-	7.66	0.63	834.0	12.58	0.884	0.006
7 d	25.59	1.87	1.89	0.31	35.31	2.88	0.849	0.031	0.63	0.09	1058.1	27.21	0.731	0.033
14 d	27.65	2.31	1.18	0.35	28.72	3.11	0.869	0.022	0.62	0.11	579.1	17.35	0.757	0.029
21 d	28.64	1.87	1.18	0.24	24.28	2.55	0.822	0.034	0.10	0.04	458.5	15.22	0.744	0.019
28 d	31.33	2.33	1.06	0.21	23.39	1.25	0.850	0.028	0.10	0.06	520.7	19.36	0.781	0.022
35 d	26.19	1.55	0.99	0.18	15.18	2.87	0.883	0.033	0.12	0.08	497.5	13.56	0.780	0.018
42 d	29.78	2.53	0.88	0.16	14.36	1.21	0.894	0.043	0.15	0.07	631.3	20.68	0.773	0.035
**24Mn3Cr**														
	** *R_s_* ** **(Ω·cm^2^)**		***Q_scale_* (×10^−4 ^** **F·cm^−2^·s^n−1^)**		** *R_scale_* ** **(Ω·cm^2^)**		** *n_scale_* **		***Q_dl_* (×10^−4 ^** **F·cm^−2^·s^n−1^)**		** *R_ct_* ** **(Ω·cm^2^)**		** *n_dl_* **	
**Time**	**μ**	**σ**	**μ**	**σ**	**μ**	**σ**	**μ**	**σ**	**μ**	**σ**	**μ**	**σ**	**μ**	**σ**
1 h	30.74	0.18	-	-	-	-	-	-	4.10	0.11	162.3	2.69	0.722	0.005
1 d	29.84	1.25	-	-	-	-	-	-	9.82	0.21	139.1	3.11	0.837	0.006
7 d	24.84	1.08	6.56	0.39	108.2	8.91	0.871	0.028	3.61	0.42	199.9	8.43	0.941	0.008
14 d	28.13	2.11	8.50	0.34	354.8	11.21	0.852	0.030	0.24	0.08	103.3	7.14	0.901	0.006
21 d	24.90	3.81	7.93	0.25	394.8	21.22	0.886	0.041	0.26	0.07	134.1	9.78	0.93	0.011
28 d	25.31	2.87	7.06	0.21	546.1	23.58	0.883	0.023	0.21	0.05	162.2	6.74	0.93	0.018
35 d	27.42	6.15	4.65	0.54	737.8	28.28	0.892	0.034	0.18	0.03	253.1	11.81	0.88	0.023
42 d	29.70	4.25	6.25	0.29	756.1	18.21	0.898	0.014	0.29	0.06	279.7	16.87	0.99	0.016

**Table 4 materials-15-01746-t004:** EDS quantitative analysis at each point marked on Figure 5.

Elements	APIX70		9Ni		24Mn3Cr	
	wt%	at%	wt%	at%	wt%	at%
O	0.17	23.69	3.20	10.44	21.48	48.31
Cr	-	-	-	-	22.52	15.58
Mn	-	-	-	-	2.42	1.59
Ni	-	-	19.76	17.56	-	-
Fe	91.83	76.31	77.04	72.00	53.58	34.52

**Table 5 materials-15-01746-t005:** EDS quantitative analysis at each point marked on Figure 6.

Elements	APIX70		9Ni		24Mn3Cr	
	wt%	at%	wt%	at%	wt%	at%
O	14.16	27.75	5.66	17.54	17.71	42.24
Cr	-	-	-	-	29.76	21.84
Mn	-	-	-	-	2.01	1.39
Ni	-	-	28.77	24.28	-	-
Fe	85.84	72.25	65.57	58.18	50.52	34.52

## Data Availability

The data supporting the findings of this study are available from the corresponding author upon reasonable request.

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
