# Peer review of "Long-Term Corrosion Behavior of Strong and Ductile High Mn-Low Cr Steel in Acidic Aqueous Environments"

_materials, 2022, doi:10.3390/ma15051746_

Round 1

Reviewer 1 Report

Reviewer Recommendation and Comments for manuscript materials-1552219 with the title: “Long-term corrosion behavior of strong and ductile high Mn-low Cr steel in acidic aqueous environments”, authors: J. S. Park, S. O. Kim, Y. J. Jeong, S. G. Lee, J. K. Choi, S. J. Kim*.

The authors study the corrosion behavior of FCC 24Mn3Cr steel in acidic aqueous conditions. This study is performed in comparison with the corrosion behavior of two BCC type steels, namely 9Ni and APIX70. Electrochemical methods such as open circuit potential (OCP), linear polarization (LP) and electrochemical impedance spectroscopy (EIS) are used to determine the electrochemical parameters of 24Mn3Cr steel over a period of 42 days. Surface morphology and structural composition are studied with surface analysis techniques such as SEM and EDS.

The topic of study in this manuscript corresponds to the subject of the Materials journal. The article may be published after revision.

The main comments that I find useful for improving the quality of the article are presented below:

*General remark. The Abstract should also include the methods used in the study. It would be appropriate for this part of the manuscript to be a little more developed.

*General remark. The "Introduction" needs to be developed in more detail. More new references need to be introduced. Also in this section, the authors should motivate the study and emphasize the practical importance of the research.

*General remark. Icorr for a sample, at different times, has been determined on the same sample or on different samples? These details should be added in the "Experimental" section.

*General remark. The study is poorly documented (only 23). Only 30% of the references are in the last 5 years (2017-2021). The study also shows very old references from 1983 (84 years old). The study needs to be supplemented with new references from recent years.

*General remark. Please re-evaluate your previous study. Why is the carbon content for 9Ni and APIX70 different from the current manuscript? (Strong and ductile Fe-24Mn-3Cr alloy resistant against erosion-corrosion, Yeong Jae Jeong, Si On Kim, Jin Sung Park, Jae Won Lee, Joong-Ki Hwang, Soon Gi Lee, Jong Kyo Choi, Sung Jin Kim, npj Materials Degradation (2021) 5:47; https://doi.org/10.1038/s41529-021-00195-0).

*line 61. Why was acetic acid and not hydrochloric acid added? The reason for this choice must be added.

*line 110. The process of reducing molecular oxygen is considered to be the main cathodic process. Why? In acidic conditions the main cathodic process is the reduction of hydrogen ions !? or not? New comments need to be made to elucidate the cathodic process.

*line 125. Figures 2a,b,c. The X axis is i or logi? The X-axis name must be changed in the logi.

*line 179. Figures 4d,e,f,g,h,i. Why Resin?

*line 180. Figure 4 must also contain the FE-SEM images of the samples, before corrosion.

*line 189. Figures 5d,e,f,g,h,i. Why Resin?

*General remark. The authors need to explain the EDS results in more detail. According to Tables 2 and 3, EDS analysis does not indicate the presence of C or Cl. Knowing that corrosive anions are chloride and acetate anions, it must be explained why there are no traces of them in corrosion products. The presence of chlorine in the form of oxychlorinated products has been proven on the surface of steels.

*The typos must be corrected.

*I think there is confusion between Funding and Acknowledgments. It needs to be checked and possibly corrected.

*The authors must revise the entire manuscript.

Author Response

The authors appreciate the valuable comments on our work. Some parts were revised, and the required information is included as advised by the reviewers. The revised parts are written in blue ink for your easy check-up in the revised manuscript.

Reviewer 2 Report

The manuscript presents a study about the corrosion behaviour, by linear polarization and electrochemical impedance spectroscopy, of 24Mn3Cr steel. Also, the sample surface was studied by SEM and EDS. However, the paper needs major revisions before it is processed further, some comments follow:

Abstract

The abstract must be improved. There is no clear purpose of the study in the abstract. Also, please highlight the novelty of the study. The abstract must be reformulated. The abstract must contain information about:

  • Background: Place the question addressed in a broad context and highlight the purpose of the study;
  • Methods: Describe briefly the main methods used to obtain and characterize the material (LP, EIS, SEM etc.).
  • Results and conclusions: Indicate the main conclusions or interpretations.

Introduction section

The introduction section must be improved.

The introduction section is too short. Please add more previous works, and highlight the experiments and results published previously.

Experimental

Please change the name of this section in Materials and Methods

Subsection 2.1. Please introduce the information about the dimensions and the shape of the tested samples. Also, please move the preparation process (lines 58-60) in this subsection.

Subsection 2.3. Please specify the soft used to obtain the equivalent circuits.

Results and discussion

In my opinion, it is not necessary a supplementary file.

Please introduce a table with the parameters measured or calculated for all three samples (Polarization resistance, corrosion potential, current density, corrosion rate etc.) and discuss them.

Also, for EIS tests please introduce a table with the following data: Rs, Rscale, CPE, εz etc. See example: DOI: 10.3390/ma14010188

The results obtained by LPR and EIS tests have been presented with very limited discussions. Please improve.

Author Response

(The authors gave the same response as above.)

Reviewer 3 Report

This work explored the long-term corrosion behavior of strong and ductile high Mn-low Cr steel in acidic aqueous environments.The long-term corrosion behavior of strong and ductile 24Mn3Cr steel under acidic aqueous conditions were examined by comparison with conventional ferritic steels. This is a good paper, but there are the following errors that need to be corrected.

  1. The language in the manuscript needs to be revised accordingly.
  2. Recertification check abbreviations throughout the manuscript.
  3. The Abstract part needs to write the innovation of the work.
  4. Introduction needs to be greatly improved. it does not reflect the novelty of the work. 5. The experimental data of the electrochemical impedance part needs to be greatly improved. The following references can be cited. Journal of Colloid and Interface Science 570 (2020) 116–124,Journal of Colloid and Interface Science 609 (2022) 838–851, Materials 2020, 13(6), 1480.
  5. The rationale for the choice of equivalent circuit diagram should be given in the manuscript.
  6. The Nyquist diagrams in Figs. 3 should give the experimental data and fitted data.
  7. The corrosion mechanism is very important, and a section should be added to describe the corrosion mechanism and give a schematic diagram of the corrosion mechanism.
  8. In the conclusion part, the experimental results should be more fully summarized.

Author Response

(The authors gave the same response as above.)

Round 2

Reviewer 1 Report

Reviewer Recommendation and Comments for manuscript materials-1552219 with the title: “Long-term corrosion behavior of strong and ductile high Mn-low Cr steel in acidic aqueous environments”, authors: J. S. Park, S. O. Kim, Y. J. Jeong, S. G. Lee, J. K. Choi, S. J. Kim*.

The authors took into account the comments of the reviewers. Consequently, they revised the manuscript, took into account the comments of the references and sent a much improved form. On this occasion I would like to congratulate the authors for their work.

More attention should be paid to typos; for example, “All authors have agreed to the published version of the manuscript. All authors have read and agreed to the published version of the manuscript. ”

Author Response

The authors appreciate the valuable comments on our work.

As advised, typos (in Author Contributions) were corrected in the revised manuscript.

Reviewer 2 Report

The authors addressed most of my comments and suggestions and the manuscript was improved accordingly. 

The paper can be accepted as it is.

Author Response

The authors appreciate the valuable comments on our work. 

Reviewer 3 Report

I have read the revised version of the Manuscript and found that the authors have taken into accounts the concerns that I raised. Thus I recommend it for publication.

Author Response

(The authors gave the same response as above.)
